# Repression of viral gene expression and replication by the unfolded protein response effector XBP1u

**Florian Hinte[1], Eelco van Anken[2,3], Boaz Tirosh[4], Wolfram Brune[1]***

[1]Heinrich Pette Institute, Leibniz Institute for Experimental Virology, Hamburg, Germany; [2]Division of Genetics and Cell Biology, San Raffaele Scientific Institute, Milan, Italy; [3]Università Vita-Salute San Raffaele, Milan, Italy; [4]Institute for Drug Research, School of Pharmacy, Faculty of Medicine, The Hebrew University, Jerusalem, Israel

**Abstract** The unfolded protein response (UPR) is a cellular homeostatic circuit regulating protein synthesis and processing in the ER by three ER-to-nucleus signaling pathways. One pathway is triggered by the inositol-requiring enzyme 1 (IRE1), which splices the X-box binding protein 1 (*Xbp1*) mRNA, thereby enabling expression of XBP1s. Another UPR pathway activates the activating transcription factor 6 (ATF6). Here we show that murine cytomegalovirus (MCMV), a prototypic β-herpesvirus, harnesses the UPR to regulate its own life cycle. MCMV activates the IRE1-XBP1 pathway early post infection to relieve repression by XBP1u, the product of the unspliced *Xbp1* mRNA. XBP1u inhibits viral gene expression and replication by blocking the activation of the viral major immediate-early promoter by XBP1s and ATF6. These findings reveal a redundant function of XBP1s and ATF6 as activators of the viral life cycle, and an unexpected role of XBP1u as a potent repressor of both XBP1s and ATF6-mediated activation.

**\*For correspondence:**
wolfram.brune@leibniz-hpi.de

## Introduction

The endoplasmic reticulum (ER) is responsible for synthesis, posttranslational modification, and folding of a substantial portion of cellular proteins. When protein synthesis is increased or ER function is compromised, the folding capacity of the ER may get out of balance, leading to an accumulation of unfolded or misfolded proteins in the ER. To alleviate ER stress and restore homeostasis, the cell activates three ER-to-nucleus signaling pathways, collectively called the unfolded protein response (UPR), which lead to a reduced protein synthesis and an increased expression of folding chaperones and ER-associated degradation (ERAD) factors (*Walter and Ron, 2011*). Subsequently, ER folding capacity increases and terminally misfolded protein species are exported from the ER and targeted for proteasomal degradation (*Christianson and Ye, 2014*).

In mammalian cells, the UPR comprises three main signaling pathways named after the initiating ER stress sensors: PERK (PKR-like ER kinase), ATF6 (activating transcription factor 6), and IRE1 (inositol-requiring enzyme 1) (*Walter and Ron, 2011*). Upon activation by ER stress, PERK phosphorylates the translation initiation factor eIF2α, which leads to a massive attenuation of protein synthesis and an immediate reduction of the protein load in the secretory system. However, phosphorylated eIF2α selectively supports the translation of selected cellular proteins such as the transcription factor ATF4, which activates a negative feedback loop resulting in dephosphorylation of eIF2α (*Novoa et al., 2001*).

Upon activation by ER stress, ATF6 travels to the Golgi, where it undergoes intramembrane proteolysis. This process liberates its cytosolic N-terminus, the basic leucine zipper (bZIP) transcription

**eLife digest** Cells survive by making many different proteins that each carry out specific tasks. To work correctly, each protein must be made and then folded into the right shape. Cells carefully monitor protein folding because unfolded proteins can compromise their viability. A protein called XBP1 is important in controlling how cells respond to unfolded proteins. Normally, cells contain a form of this protein called XBP1u, while increasing numbers of unfolded proteins trigger production of a form called XBP1s. The change from one form to the other is activated by a protein called IRE1.

Viruses often manipulate stress responses like the unfolded protein response to help take control of the cell and produce more copies of the virus. Murine cytomegalovirus, which is known as MCMV for short, is a herpes-like virus that infects mice; it stops IRE1 activation and XBP1s production during the later stages of infection. However, research had shown that the unfolded protein response was triggered for a short time at an early stage of infection with MCMV, and it was unclear why this might be.

Hinte et al. studied the effect of MCMV on cells grown in the laboratory. The experiments showed that a small dose of cell stress, namely activating the unfolded protein response briefly during early infection, helps to activate genes from the virus that allow it to take over the cell. Together, XBP1s and another protein called ATF6 help to switch on the viral genes. The virus also triggers IRE1 helping to reduce the levels of XBP1u, which could slow down the infection. Later, suppressing the unfolded protein response allows copies of the virus to be made faster to help spread the infection.

These findings reveal new details of how viruses precisely manipulate their host cells at different stages of infection. These insights could lead to new ways to manage or prevent viral infections.

factor ATF6(N), and enables it to travel to the nucleus, where it activates the transcription of chaperone genes as well as of the gene encoding XBP1 (*Lee et al., 2002*).

The third sensor, IRE1 (also known as ER-to-nucleus signaling 1, ERN1), is an ER transmembrane protein kinase that oligomerizes upon accumulation of unfolded proteins in the ER lumen. Oligomerization and auto-transphosphorylation activate the RNase function of IRE1, which mediates an unconventional splicing of the XBP1 mRNA in the cytosol (*Calfon et al., 2002*; *Lee et al., 2002*; *Yoshida et al., 2001*). Removal of the 26-nt intron from the XBP1 mRNA leads to a frame shift and expression of transcription factor XBP1s, comprising an N-terminal basic leucine zipper (bZIP) domain followed by a C-terminal transcription activation domain. By contrast, the unspliced XBP1 mRNA encodes XBP1u, which lacks the transcription activation domain but contains in its C-terminal part a hydrophobic patch and a translational pausing region required for the recruitment of its own mRNA to the ER membrane (*Kanda et al., 2016*; *Yanagitani et al., 2011*). XBP1u is rapidly degraded and has a short half-life (*Tirosh et al., 2006*). It can interact with XBP1s and ATF6(N) and target them for proteasomal degradation. Therefore, XBP1u is thought to act as a negative regulator involved in fine-tuning the UPR (*Tirosh et al., 2006*; *Yoshida et al., 2006*; *Yoshida et al., 2009*). Moreover, XBP1u affects autophagy by interacting with transcription factor FOXO1 (*Zhao et al., 2013*). Apart from mediating XBP1 mRNA splicing, IRE1 can also cleave ER-associated mRNA molecules that contain a specific recognition motif (*Moore and Hollien, 2015*). This process, which leads to mRNA degradation, is called regulated IRE1-dependent mRNA decay (RIDD). However, the importance of RIDD in different cellular processes such as lipid metabolism, antigen presentation, and apoptosis remains incompletely understood (*Maurel et al., 2014*).

During viral replication large quantities of viral proteins must be synthesized. Folding, maturation, and posttranslational modification of secreted and transmembrane proteins take place in the ER and require a plethora of chaperones, foldases, and glycosylating enzymes. While properly folded proteins are transported to the Golgi, unfolded or misfolded proteins are retained in the ER and exported to the cytosol for proteasomal degradation via the ERAD pathway (*Smith et al., 2011*). However, the high levels of viral envelope glycoproteins that are being synthesized particularly during the late phase of the viral life cycle can overwhelm the folding and processing capacity of the ER and cause accumulation of unfolded and misfolded proteins in the ER (*Zhang and Wang, 2012*).

Cytomegaloviruses (CMVs) are prototypic members of the β subfamily of the *Herpesviridae*. Their large double-stranded DNA genomes contain at least 165 protein-coding ORFs (*Dolan et al., 2004*) end encode an even larger number of polypeptides (*Stern-Ginossar et al., 2012*). Through millions of years of co-evolution with their respective hosts, the CMVs have acquired the ability to moderate immune recognition and modulate cellular stress responses to their own benefit (*Alwine, 2008*; *Mocarski, 2002*). Considering the important role of the UPR in controlling cell fitness, it is hardly surprising that the CMVs have evolved means to modify the UPR. For instance, human and murine CMV (HCMV and MCMV) induce PERK activation, but limit eIF2α phosphorylation. By doing this the CMVs prevent a global protein synthesis shutoff but allow eIF2α phosphorylation-dependent activation of transcription factor ATF4 (*Isler et al., 2005*; *Qian et al., 2012*). The CMVs also increase expression of the ER chaperone BiP to facilitate protein folding and virion assembly (*Buchkovich et al., 2008*; *Buchkovich et al., 2010*; *Qian et al., 2012*), and HCMV uses PERK to induce lipogenesis by activating the cleavage of sterol regulatory element binding protein 1 (*Yu et al., 2013*). We have previously shown that both, MCMV and HCMV, downregulate IRE1 levels and inhibit IRE1 signaling at late times post infection. This downregulation is mediated by the viral proteins M50 and UL50, respectively (*Stahl et al., 2013*). However, a real-time transcriptional profiling study has revealed that cellular ER stress response transcripts are upregulated as early 5–6 hr after MCMV infection (*Marcinowski et al., 2012*).

Here, we show that MCMV transiently activates the IRE1-XBP1 pathway at early times post infection to relieve repression of viral gene expression and replication by XBP1u. When IRE1-mediated XBP1 mRNA splicing is inhibited, XBP1u blocks the activation of the viral major immediate-early promoter (MIEP) by XBP1s and ATF6(N). Thus, MCMV exploits UPR signaling to boost the activity of its most important promoter. Moreover, these findings reveal a redundant function of XBP1s and ATF6 as activators of viral gene expression and replication, and an unexpected role of XBP1u as a potent repressor of both XBP1s and ATF6-mediated activation.

## Results

### Early activation of IRE1-XBP1 signaling promotes MCMV replication

Previous studies have shown that MCMV inhibits IRE1-XBP1 signaling at late times (≥24 hr) post infection (*Qian et al., 2012*; *Stahl et al., 2013*). However, cellular ER stress response transcripts were shown to be upregulated at 5–6 hr after MCMV infection (*Marcinowski et al., 2012*), suggesting that UPR signaling is activated at early times post infection. Thus, we decided to analyze whether MCMV activates the IRE1-XPB1 signaling pathway within the first few hours after infection. To do this, we infected mouse embryonic fibroblasts (MEFs) with MCMV and quantified spliced and unspliced XBP1 transcripts by qRT-PCR. We detected a short and transient increase of XBP1 splicing between 5 and 7 hr post infection (hpi) (*Figure 1A*). This increase was massively reduced when cells were infected with UV-inactivated MCMV (*Figure 1A*), suggesting that XBP1 splicing was not caused by viral attachment and entry into cells but required viral gene expression. The transient activation of the IRE1-XBP1 pathway was confirmed by immunoblot detection of phosphorylated IRE1 and XBP1s (*Figure 1B*). MCMV-induced XBP1 splicing was suppressed by cycloheximide (CHX, a translation inhibitor), but not by phosphonoacetic acid (PAA), an inhibitor of viral DNA replication and late gene expression (*Figure 1C*). These results suggested that the transient activation of the IRE1-XBP1 pathway is caused by viral proteins expressed at immediate-early or early times post infection.

To determine whether IRE1 signaling is important for the MCMV life cycle, we used IRE1-deficient (*Ern1*[-/-]) cells expressing IRE1-GFP under tight control of a tetracycline-inducible promoter (TetON-IRE1-GFP cells, *Figure 2A*) for analyses of viral replication. IRE1-GFP expression was induced with different concentrations of doxycycline, and cells were infected at low or high multiplicity of infection (MOI) for multi-step and single-step growth curves, respectively (*Figure 2B and C*). In both types of replication analysis, MCMV replicated to low titers when IRE1 expression was induced with very low or very high doxycycline concentrations. High MCMV titers (~$10^6$ infectious units per ml), comparable to those obtained in wildtype (WT) MEFs, were attained only upon moderate induction of IRE1-GFP with 5 to 10 nM doxycycline (*Figure 2B and C*). We observed that IRE1-GFP induction with high doxycycline concentrations resulted in a significantly decreased cell viability (*Figure 2D*), suggesting that IRE1 overexpression is detrimental to cell viability. This is consistent with the previously

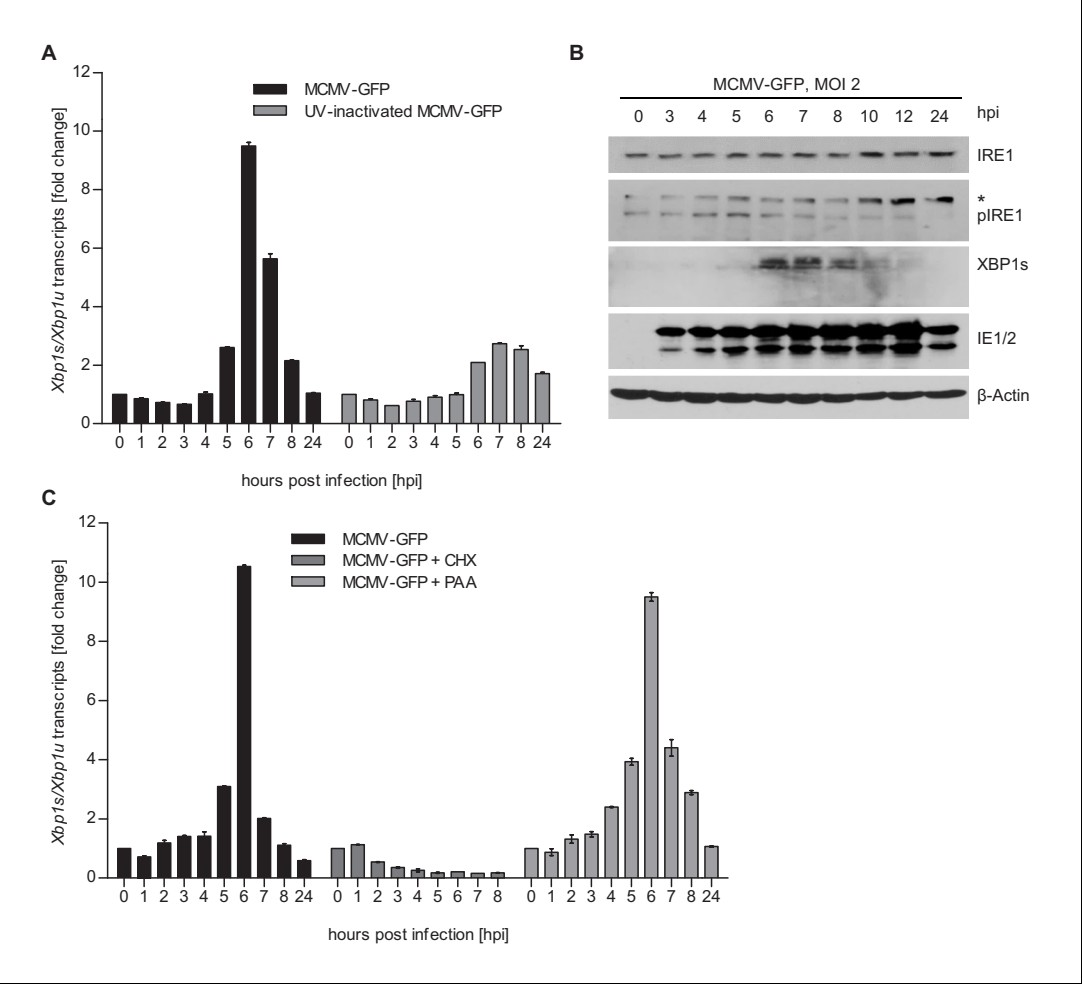

**Figure 1.** MCMV induces *Xbp1s* mRNA splicing at early time of infection. (**A**) MEFs were infected with MCMV-GFP or UV-inactivated MCMV-GFP (MOI 4). Cells were harvested at the indicated times, total RNA was extracted, and *Xbp1s* and *Xbp1u* transcripts were quantified by qPCR. Changes in the *Xbp1s/Xbp1u* ratio relative to uninfected cells are plotted as bar diagram (means ± SEM of 3 biological replicates). (**B**) Immunoblot analysis of MEFs infected with MCMV-GFP. Endogenous IRE1, phosphorylated IRE1, and XBP1s were detected using specific antibodies. *, unspecific band. The immunoblot is representative of 2 independent experiments. (**C**) MEFs were infected with MCMV-GFP as described above and treated with vector, CHX (50 µg/ml) or PAA (250 ng/ml). Changes in the *Xbp1s/Xbp1u* ratio were determined as described above. Data provided in *Figure 1—source data 1*. The online version of this article includes the following source data for figure 1:

**Source data 1.** Data points of qRT-PCR.

reported cytotoxicity of IRE1 overexpression (*Han et al., 2009*). To formally exclude the possibility that strong activation of the TetON transactivator alone impairs MCMV replication, we infected TetON-expressing cells with MCMV and analyzed viral replication in the presence of different doxycycline concentrations. As expected, MCMV replication was not affected by different doxycycline concentration (*Figure 2—figure supplement 1*). Thus, we concluded that IRE1 is necessary for efficient replication of MCMV, but needs to be carefully regulated as too high expression levels are detrimental for cell viability and viral replication.

## IRE1, but not XBP1 or TRAF2, is required for efficient MCMV gene expression and replication

Activated IRE1 can splice *Xbp1* mRNA (*Calfon et al., 2002*; *Lee et al., 2002*; *Yoshida et al., 2001*) and can also recruit TRAF2 to activate ASK1 (*Urano et al., 2000*). To test which IRE1-dependent signaling pathway is required for efficient MCMV replication, we used CRISPR/Cas9-mediated gene editing to generate knockout (ko) MEFs for *Ern1* (the gene encoding IRE1), *Xbp1*, and *Traf2*. For

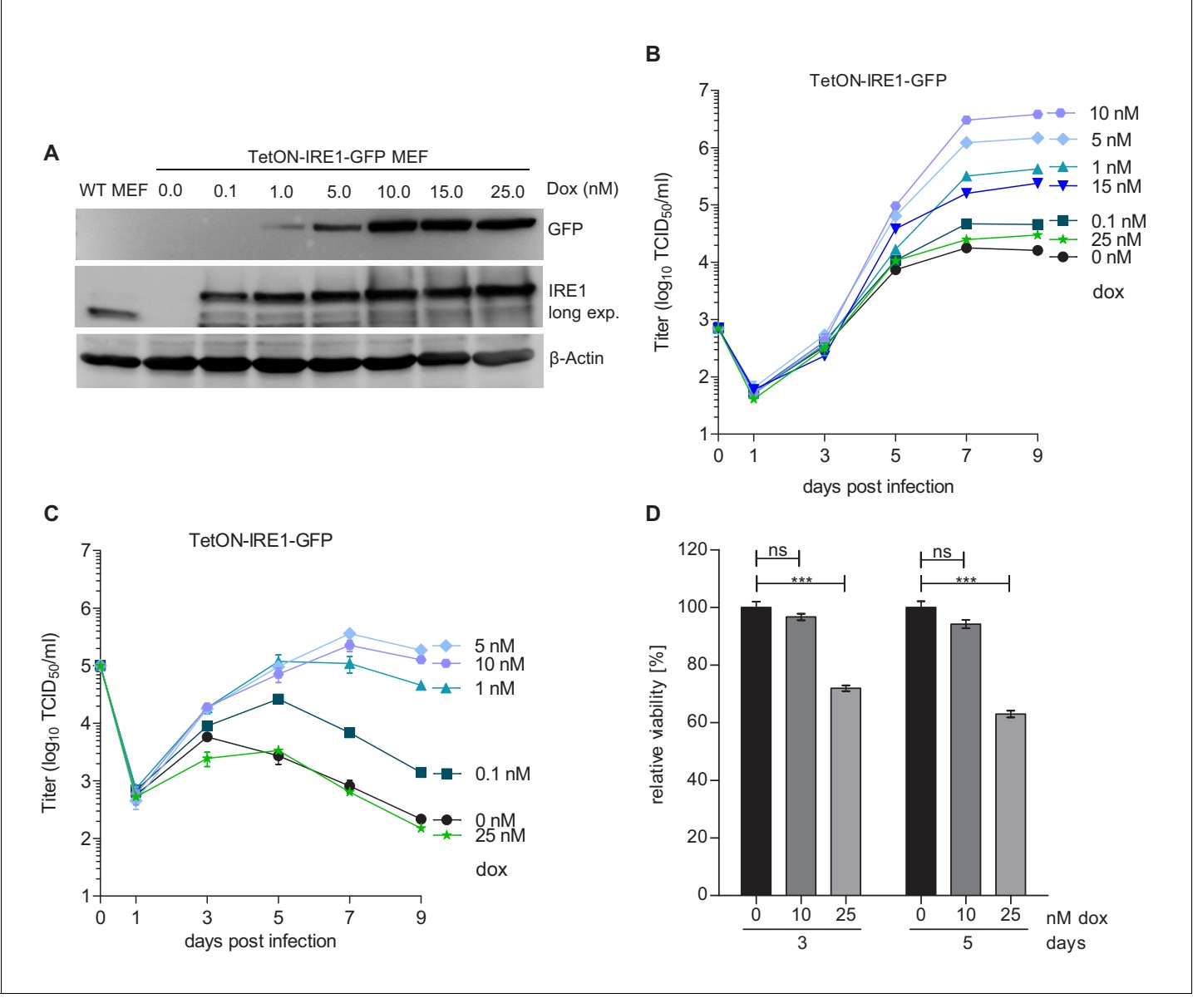

**Figure 2.** Moderate IRE1 expression is beneficial for MCMV replication. (**A**) Immunoblot analysis of IRE1-deficient (*Ern1*[-/-]) MEFs expressing IRE1-GFP (TetON-IRE1-GFP) in a doxycycline (dox)-inducible manner. Cells were treated with different dox concentrations for 24 hr. IRE1-GFP expression was detected with GFP or IRE1-specific antibodies. Endogenous IRE1 levels in WT MEFs were detected only with the IRE1-specific antibody. The immunoblot is representative of 3 independent experiments. (**B**) Multistep MCMV replication kinetics on TetON-IRE1-GFP MEFs induced with different dox concentrations. 24 hr after induction, cells were infected with MCMV-GFP (MOI 0.1). Virus titers in the supernatants were determined by titration and are shown as means ± SEM of 3 biological replicates. (**C**) Single step MCMV replication kinetics on TetON-IRE1-GFP MEFs induced with dox as above, infected with MCMV-GFP (MOI 3) and are shown as means ± SEM of 3 biological replicates. (**D**) Cell viability of TetON-IRE1-GFP MEFs treated with different dox concentrations. Cell viability was measured after 3 and 5 days of dox treatment and is shown as relative viability compared to untreated cells (means ± SEM of 6 biological replicates). Data provided in *Figure 2—source data 1*. Additional data provided in *Figure 2—figure supplement 1*.

The online version of this article includes the following source data and figure supplement(s) for figure 2:

**Source data 1.** Data points of growth curves.

**Figure supplement 1.** MCMV replication kinetics on TetON expressing cells.

each gene knockout, two independent cell clones were generated with different guide RNAs. The absence of the respective gene products was verified by immunoblot analysis (*Figure 3A*). Then, the cell clones were used to assess MCMV replication. In *Ern1* ko MEFs, viral replication (*Figure 3B*) and

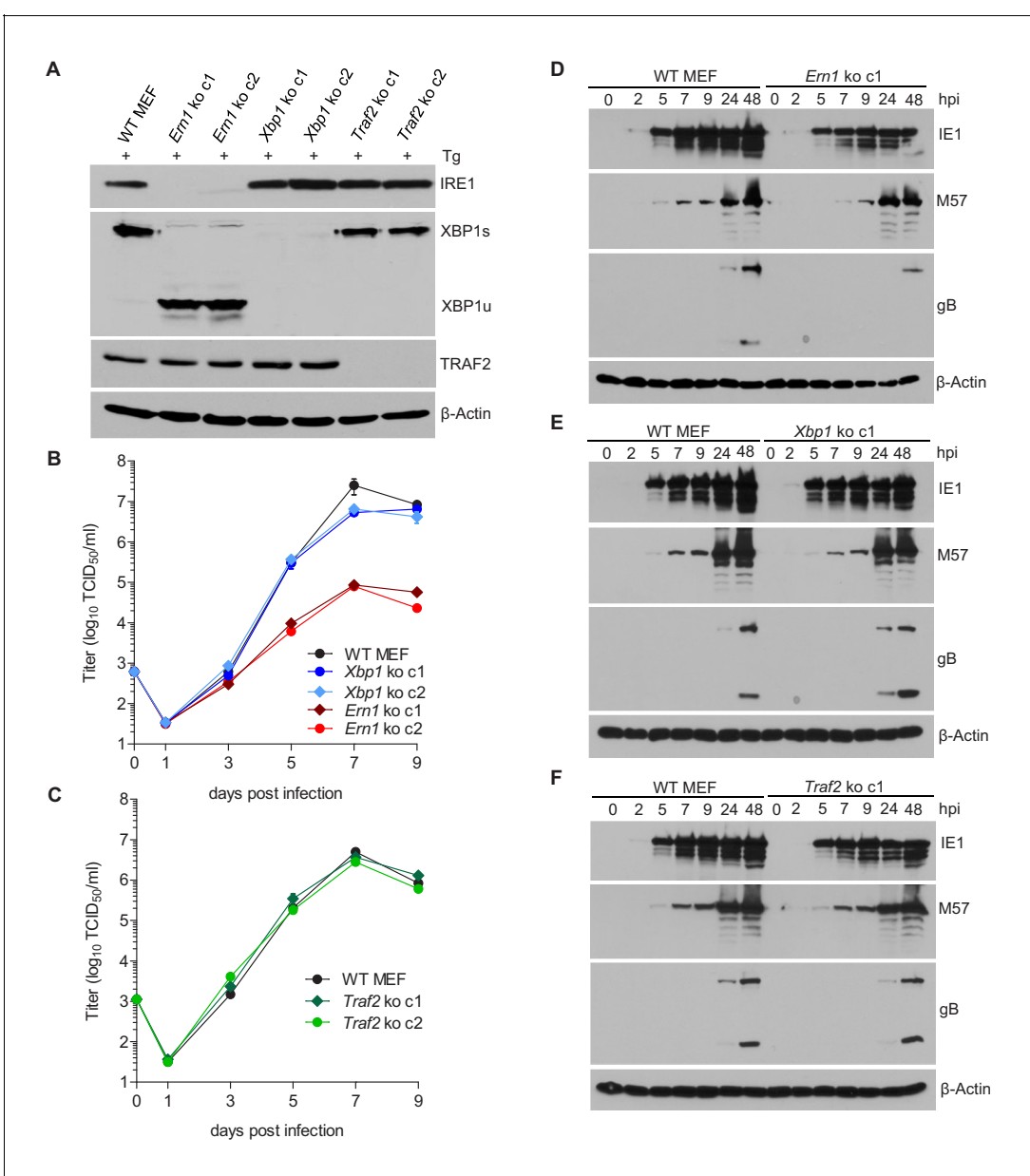

**Figure 3.** IRE1, but not XBP1 or TRAF2, is required for efficient MCMV replication and viral protein expression. (**A**) Immunoblot analysis of IRE1, XBP1, and TRAF2-deficient (*Ern1*, *Xbp1*, and *Traf2* ko) cell lines. Two ko clones were generated for each gene by CRISPR/Cas9 gene editing using different gRNAs. Cells were treated for 4 hr with Thapsigargin (Tg) to induce *Xbp1* mRNA splicing and to increase XBP1 expression. (**B,C**) Multistep MCMV replication kinetics in *Ern1*, *Xbp1* and *Traf2* ko cells, respectively. Cells were infected with MCMV-GFP (MOI 0.1). Virus titers in the supernatants were determined by titration and are shown as means ± SEM of 3 biological replicates. (**D–F**) Immunoblot analysis of viral protein expression kinetics in *Ern1*, *Xbp1* and *Traf2* ko cells, respectively. Cells were infected with MCMV-GFP (MOI 3) and harvested at different times post infection. Expression levels of the viral immediate-early 1 (IE1) protein, the major DNA binding protein (M57; an early protein), and glycoprotein B (gB; a late protein) were detected with specific antibodies, β-Actin served as loading control. Immunoblots are representative of 2 independent experiments. Data provided in *Figure 3—source data 1*. Additional data provided in *Figure 3—figure supplement 1*.

The online version of this article includes the following source data and figure supplement(s) for figure 3:

**Source data 1.** Data points of growth curves and qRT-PCR.

**Figure supplement 1.** qRT-PCR analysis of viral transcripts in WT and IRE1-deficient cells.

viral gene transcription (*Figure 3—figure supplement 1*) were massively reduced as compared to WT MEFs (*Figure 3B*), similar to the reduction seen in IRE1-GFP cells without doxycycline induction (*Figure 2B*). By contrast, MCMV replication was virtually unimpaired in the absence of *Xbp1* (*Figure 3B*) or *Traf2* (*Figure 3C*). We also analyzed the expression of a viral immediate-early (IE1), an early (M57), and a late protein (gB) at different times after high-MOI infection. Compared to WT MEFs, the expression of all three proteins was reduced in *Ern1* ko MEFs (*Figure 3D*), but not in *Xbp1* or *Traf2* ko MEFs (*Figure 3E and F*).

Next, we tested whether the RNase activity of IRE1 is required for efficient MCMV replication. To do this, WT IRE1 or an 'RNase-dead' IRE1-K907A mutant (*Tirasophon et al., 2000*) was expressed in *Ern1* ko MEFs by retroviral transduction. Expression of WT and mutant IRE1 and the ability to splice *Xbp1* was verified by immunoblot analysis (*Figure 4A*). While expression of WT IRE1 restored MCMV replication to high titers, expression of IRE1-K907A did not increase MCMV titers (*Figure 4B*), indicating that the IRE1 RNase activity is necessary for efficient MCMV replication.

## XBP1u inhibits MCMV replication

Our observations that the RNase activity of IRE1 is required for efficient MCMV replication, but XBP1 is not, allowed two possible explanations: (i) MCMV replication could benefit from RIDD, another RNase-dependent function of IRE1. However, this possibility is difficult to verify as selective inactivation of RIDD is complicated. (ii) Alternatively, MCMV replication could be inhibited by XBP1u, the product of the unspliced *Xbp1* mRNA, since *Ern1* ko cells differ from other cells in that they express only XBP1u (*Figure 3A*). To test the latter option, we used two experimental approaches. First, we knocked out *Ire1* in *Xbp1*-deficient cells (*Figure 5A*). As shown in *Figure 5B*, MCMV replicated to similar titers in *Xbp1* ko and *Xbp1/Ern1* double-knockout (dko) cells, indicating that the loss of IRE1 is not detrimental for MCMV replication when XBP1 is absent. Next, we used retroviral transduction to restore XBP1 expression in *Xbp1⁻/⁻* cells. The retroviral vectors expressed the WT *Xbp1* transcript (which can be spliced by IRE1), a truncated *Xbp1* transcript encoding only the DNA-binding domain (XBP1stop), or an 'unspliceable' *Xbp1* transcript. Upon treatment with thapsigargin, these cells expressed XBP1s, XBP1stop, or XBP1u, respectively (*Figure 5C*). Whereas re-introduction of WT XBP1 had no detrimental effect, MCMV replication was severely impaired upon expression of the unspliceable *Xbp1* transcript (*Figure 5D*), indicating that *Xbp1u* reduces viral

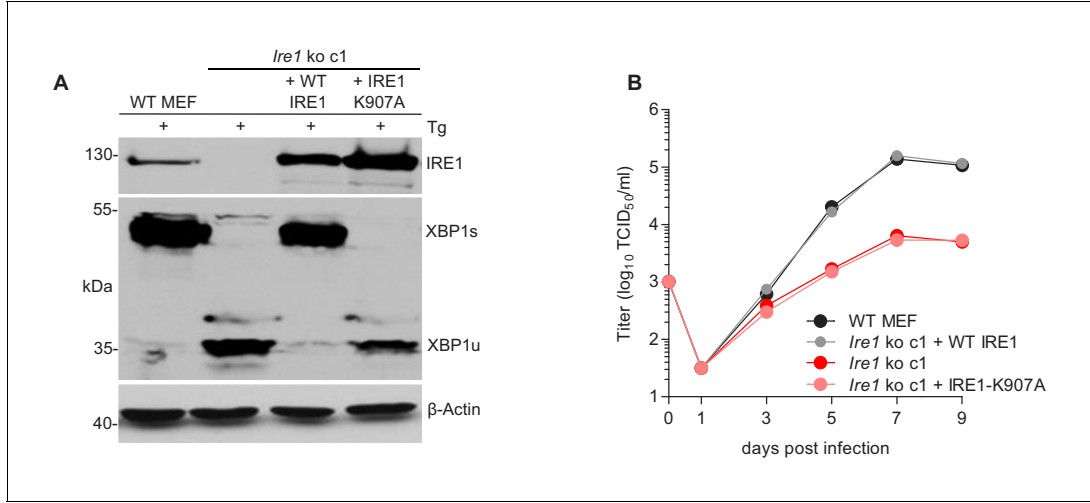

**Figure 4.** The RNase function of IRE1 is required for efficient MCMV replication. (**A**) Immunoblot analysis of IRE1-deficient cells (*Ern1* ko c1) transduced with retroviral vectors expressing WT IRE1 or IRE1-K907A (RNase-dead). Cells were treated for 4 hr with Thapsigargin (Tg) to induce *Xbp1* mRNA splicing and to increase XBP1 expression. IRE1 and XBP1 protein expression was detected by immunoblot (representative of 2 independent experiments). (**B**) Multistep MCMV replication kinetics in *Ern1 ko* cells complemented with WT IRE1 or IRE1-K907A, respectively. Cells were infected with MCMV-GFP (MOI 0.1). Virus titers in the supernatants were determined by titration and are shown as means ± SEM of 3 biological replicates. Data provided in *Figure 4—source data 1*.

The online version of this article includes the following source data for figure 4:

**Source data 1.** Data points of growth curves.

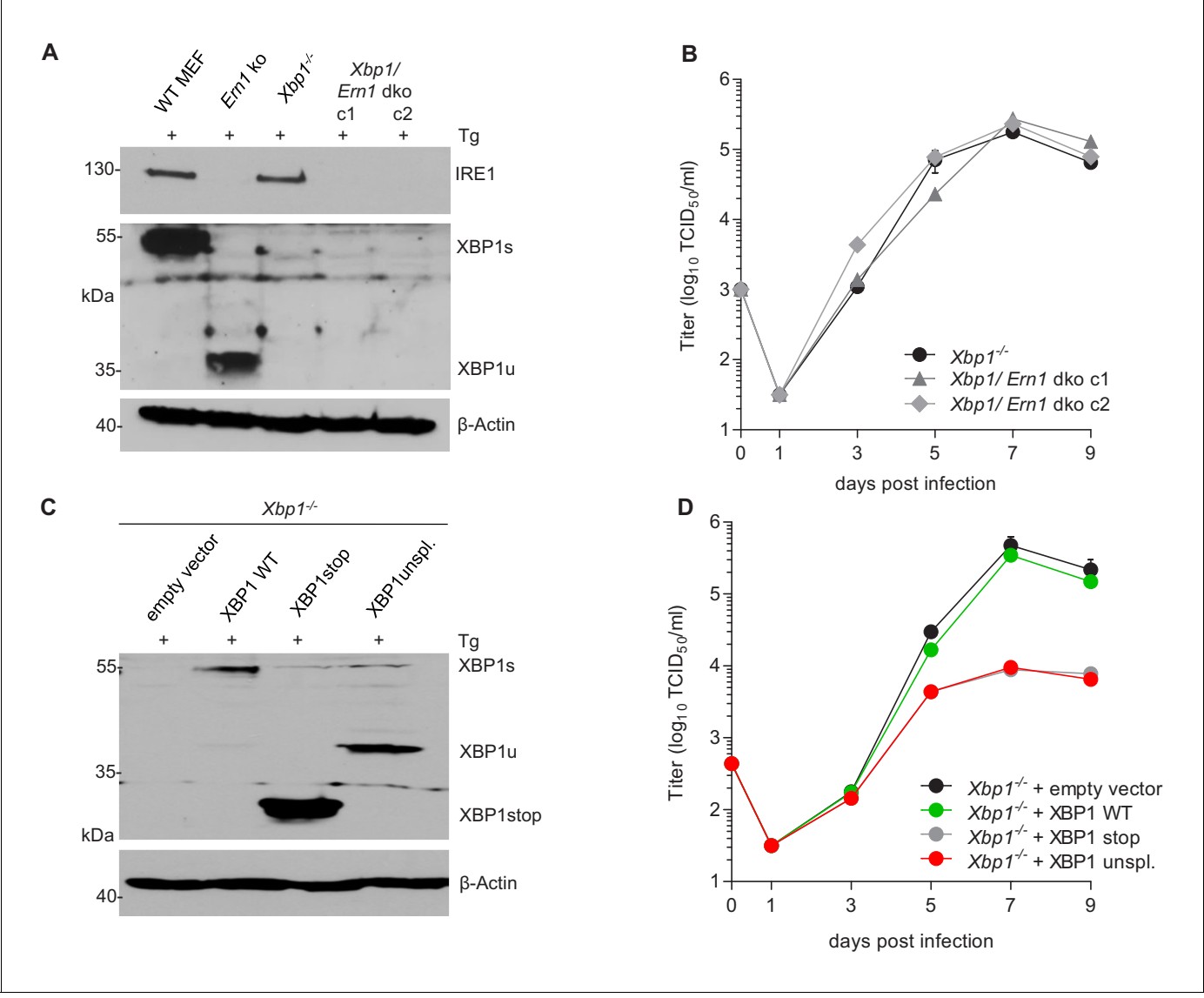

**Figure 5.** XBP1u acts as a repressor for MCMV replication. (**A**) *Xbp1*[-/-] MEFs were used to knock out the IRE1-encoding *Ern1* gene by CRISPR/Cas9 gene editing. Two double ko (dko) cell clones were generated with different gRNAs. IRE1 and XBP1 protein expression in the two dko clones and control cells was detected by immunoblot analysis. Cells were treated for 4 hr with Thapsigargin (Tg) to induce *Xbp1* mRNA splicing and to increase XBP1 expression. (**B**) Multistep MCMV replication kinetics in *Xbp1*[-/-] and *Xbp1/Ern1* dko MEFs. Cells were infected with MCMV-GFP (MOI 0.1). Virus titers in the supernatants were determined by titration and are shown as means ± SEM of 3 biological replicates. (**C**) Immunoblot analysis of *Xbp1*[-/-] MEF transduced with retroviral vectors expressing a WT (spliceable) *Xbp1* transcript, an unspliceable *Xbp1* transcript, or a truncated (*Xbp1*stop) transcript. Cells were treated with Tg as described in A. (**D**) Multistep MCMV replication kinetics in cells shown in C. Infection and titration was done as in B. Immunoblots are representative of 2 independent experiments. Data provided in *Figure 5—source data 1*.

The online version of this article includes the following source data for figure 5:

**Source data 1.** Data points of growth curves.

replication. A similar inhibitory effect was observed when the truncated XBP1 protein was expressed (*Figure 5D*).

## Loss of XBP1 and ATF6 impairs MCMV replication

XBP1u is thought to function as a negative regulator to XBP1s (*Tirosh et al., 2006*; *Yoshida et al., 2006*). However, the inhibition of MCMV replication by XBP1u cannot be explained solely by a repressive effect on XBP1s as a complete loss of XBP1 is not detrimental to MCMV replication in

vitro (*Figure 3B*) and has only a modest effect on viral replication in vivo (*Drori et al., 2014*). Thus, we hypothesized that XBP1u might impair MCMV replication by repressing the activity of additional transcription factors besides XBP1s. For several reasons ATF6 appeared to be a likely target of XBP1u: (i) Like XBP1s, ATF6 is a bZIP transcription factor activated by ER stress (*Yoshida et al., 1998*); (ii) XBP1s and ATF6 are known to synergize in the activation of numerous ER stress response genes (*Lee et al., 2002*); and (iii) XBP1u can interact with ATF6 and inhibit its activity (*Yoshida et al., 2009*). Therefore, we tested whether the loss of both, XBP1 and ATF6, was detrimental for MCMV replication. First, we analyzed MEFs from *Atf6*[-/-] mice and found that MCMV gene expression and replication were not impaired (*Figure 6A and B*). When we knocked out *Xbp1* in *Atf6*[-/-] cells by CRISPR/Cas9 gene editing (*Figure 6C*), MCMV replication was substantially reduced in *Atf6/Xbp1* dko cells (*Figure 6D*), suggesting that ATF6 and XBP1s have overlapping or redundant functions and that at least one of them is necessary for efficient MCMV replication.

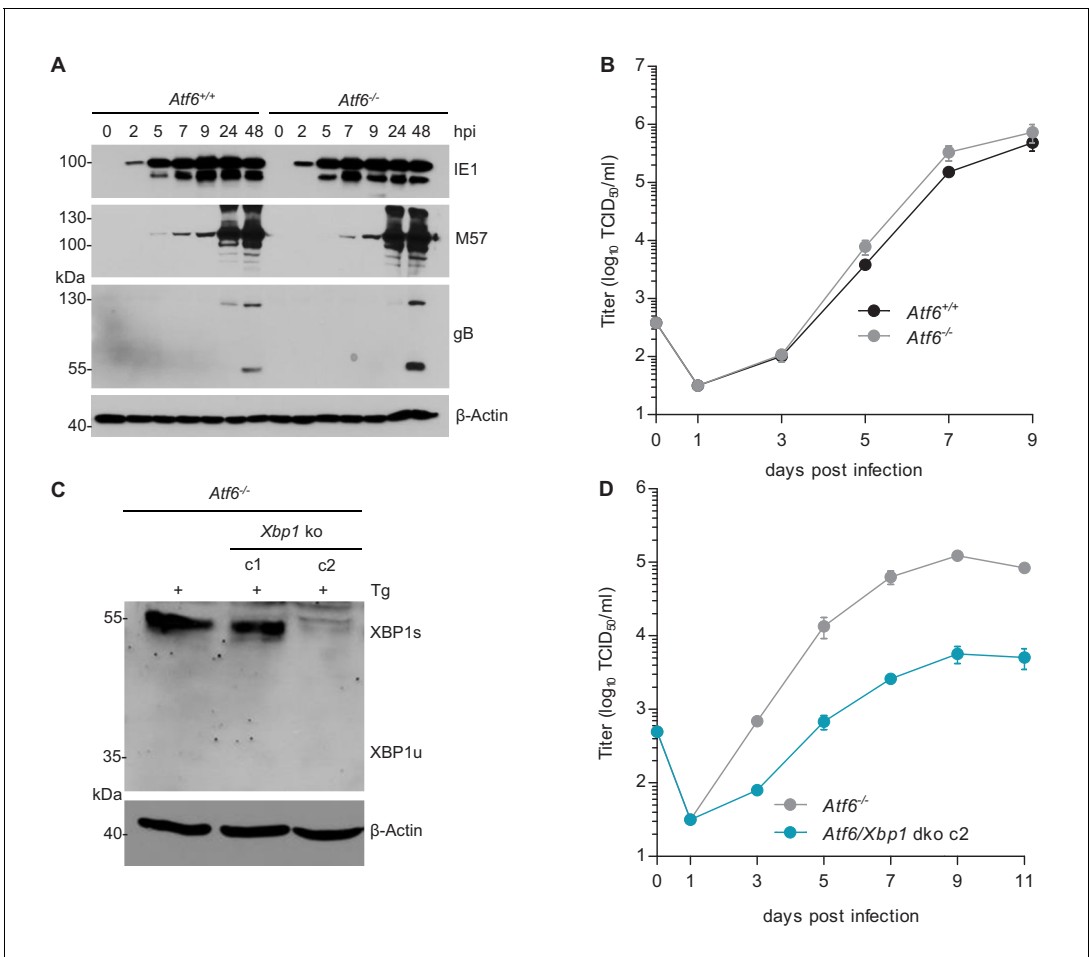

**Figure 6.** MCMV replication is impaired in cells lacking *Atf6* and *Xbp1*. (**A**) Immunoblot analysis of viral gene expression of *Atf6*[+/+] and *Atf6*[-/-] cells infected with MCMV-GFP (MOI 3). At the indicated times cells were lysed and stained for the immediate-early one protein (IE1), the major DNA binding protein (M57; early gene) and glycoprotein B (gB; late gene) by immunoblot. β-Actin served as loading control. (**B**) Multistep replication kinetics. *Atf6*[+/+] and *Atf6*[-/-] cells were infected with MCMV-GFP (MOI of 0.1). Virus titers in the supernatants were determined by titration and are shown as means ± SEM of 3 biological replicates. (**C**) Knockout of *Xbp1* in *Atf6*[-/-] cells using CRISPR/Cas9 gene editing. Two single cell clones generated by two individual gRNAs (c1 and c2) were analyzed for XBP1s and XBP1u expression by immunoblot. 4 hr prior to harvesting, cells were stimulated with thapsigargin (Tg) to enhance XBP1 expression. The parental *Atf6*[-/-] cells are shown as control. (**D**) Multistep replication kinetics of *Atf6/Xbp1* dko cells (clone c2). MCMV infection and titration was done as described in B. Immunoblots are representative of 2 independent experiments. Data provided in *Figure 6—source data 1*.

The online version of this article includes the following source data for figure 6:

**Source data 1.** Data points of growth curves.

## XBP1s and ATF6-mediated activation of the MCMV major immediate-early promoter (MIEP) is repressed by XBP1u

The MCMV MIEP has a key role in viral gene expression as it drives the expression of the viral IE1 and IE3 proteins. IE3 is the major transactivator protein that activates early and late gene expression (*Lacaze et al., 2011*). Thus we interrogated whether XBP1s and ATF6 can activate the MCMV MIEP. First, we searched for ACGT motifs within the MCMV MIEP. ACGT is a minimal consensus sequence contained within XBP1s and ATF6 binding motifs (*Kanemoto et al., 2005*; *Wang et al., 2000*). Five ACGT motifs were identified within the MCMV MIEP (*Figure 7A*). To analyze the function of these putative transcription factor binding sites, we inserted the WT MIEP and six mutant versions (*Supplementary file 1*) into the luciferase reporter plasmid pGL3-Basic. In the mutant MIEPs, one or all five ACGT motifs were changed to CTAG. Using a luciferase reporter assay, we measured MIEP activity in WT, *Ern1* ko and *Xbp1* ko fibroblasts. As shown in *Figure 7B*, MIEP activity was strongly reduced in *Ern1* ko cells but was not significantly altered in *Xbp1* ko cells. This result is consistent with the MCMV replication defect observed in *Ern1*, but not *Xbp1* ko cells (*Figure 3B*). MIEP activity was not reduced in *Xbp1/Ern1* dko cells compared to the parental *Xbp1* ko cell (*Figure 7C*), but was significantly reduced in *Atf6/Xbp1* dko cells compared to the parental *Atf6⁻/⁻* cells (*Figure 7D*). Again, the MIEP activities were consistent with the results of the viral replication kinetics (*Figures 5B* and *6D*). Thus, we concluded that activation of the MCMV MIEP correlated with viral replication in the same cells. We also found that the activity of the MIEP had a substantially reduced activity when all 5 ACGT motifs were mutated (*Figure 7B–D*).

Next, we tested whether XBP1s and ATF6(N), the active form of ATF6, can activate the MCMV MIEP and whether XBP1u can repress it. To assess the contribution of endogenous levels of the TFs, *Xbp1* ko, *Atf6⁻/⁻*, and *Xbp1/Atf6* dko cells were used. In *Xbp1* ko cells, MIEP activity was slightly increased by expression of XBP1s, but substantially reduced by XBP1u. XBP1u also antagonized XBP1s in a dose-dependent manner (*Figure 7E*). A similar result was obtained in *Atf6⁻/⁻* cells: MIEP activity was slightly increased by expression of ATF6(N), but substantially reduced by XBP1u. XBP1u also antagonized ATF6(N) in a dose-dependent manner (*Figure 7F*). In *Atf6/Xbp1* dko cell, expression of either XBP1s or ATF6(N) was sufficient to increase MIEP activity substantially. XBP1u alone did not reduce MIEP activity in these cells, but it antagonized the activity of ATF6(N) expressed by transfection (*Figure 7G*). Taken together, these results suggest that both, XBP1s and ATF6(N), can activate the MCMV MIEP in a largely redundant fashion, and that XBP1u represses the activity of both, XBP1s and ATF6(N).

## Motif 4 is necessary and sufficient for MIEP activation by XBP1s and ATF6(N)

As a first step to determine, which of the five ACGT motifs function as XBP1s and/or ATF6(N) binding sites for promoter activation, we measured the binding of these TFs to portions of the MIEP by using a DNA-protein interaction ELISA (DPI-ELISA, [*Brand et al., 2010*; *Underwood et al., 2013*]). Microtiter plates were coated with double-stranded oligonucleotides encoding three copies (27 nucleotides each) of a putative binding motif with adjacent sequences on either side (*Supplementary file 2*). A known XBP1 binding motif of the ERdj4 promoter (*Kanemoto et al., 2005*) served as positive control and a sequence of the SV40 origin of replication as negative control. The oligonucleotides were incubated with HA-tagged XBP1s, XBP1u, or ATF6(N) (*Figure 8A*), and DNA binding was quantified by ELISA. Mutated TFs lacking the DNA-binding domain (ΔDBD) served as negative controls. In this DPI-ELISA, XBP1s and XBP1u showed the strongest interaction with motifs 3 and 4, whereas ATF6(N) interacted with motifs 2 and 4 (*Figure 8B*).

Next, we used the luciferase reporter assay to test which of the five motifs was required for MIEP activation by XBP1s and ATF6(N). We used five mutant MIEPs having one of the ACGT motifs changed to CTAG. These reporter plasmids were transfected into *Atf6/Xbp1* dko cells, together with expression plasmids for XBP1s, XBP1u, and ATF6(N). As shown in *Figure 9A*, XBP1s or ATF6(N) expression increased the activity of all MIEP constructs except MIEP-4-mut and MIEP-all-mut, indicating that ACGT motif 4 is necessary for MIEP activation by XBP1s and ATF6(N).

We tested then whether motif 4 was also sufficient for MIEP activation by XBP1s and ATF6(N). Motif 4 was restored in MIEP-all-mut to generate a MIEP containing only ACGT motif 4 (MIEP-4-only, *Supplementary file 1*). Indeed, MIEP-4-only was inducible by XBP1s and ATF6(N) (*Figure 9B*),

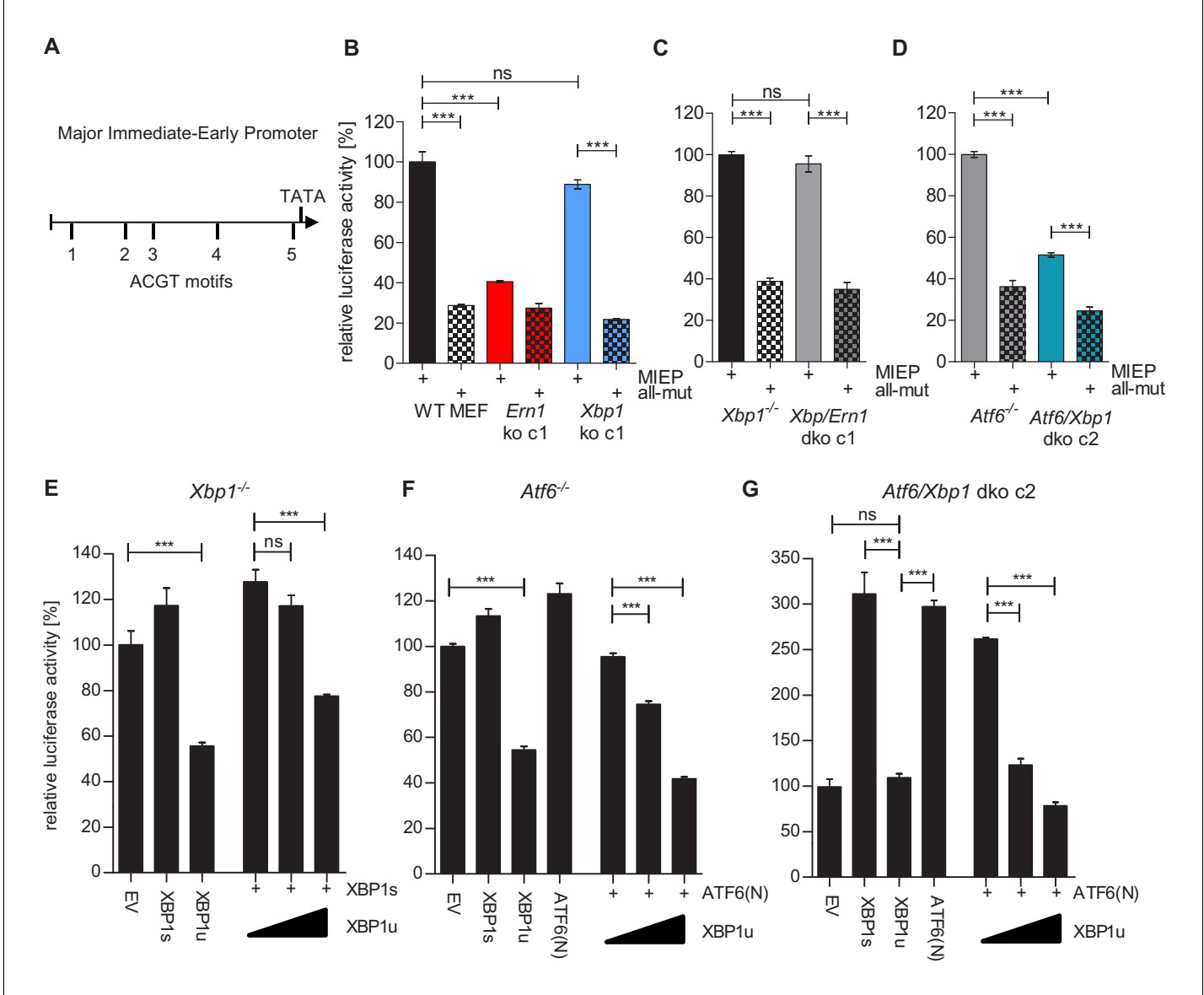

**Figure 7.** XBP1s and ATF6-mediated activation of the MCMV MIEP is repressed by XBP1u. (**A**) Schematic representation of the MCMV major immediate-early promoter (MIEP) with TATA box and 5 ACGT motifs. (**B**) WT MEFs, *Ern1* ko and *Xbp1* ko cells were transfected with a firefly luciferase vector containing either the WT MIEP or a MIEP with five mutated ACGT motifs (all-mut). Renilla luciferase was expressed by co-transfection and used for normalization. Relative luciferase activities (firefly: renilla) ± SEM of at least three biological replicates are shown. ***, p<0.001; ns, not significant, p>0.05. (**C**) *Xbp1^-/-^* and *Xbp1/Ern1* dko cells were transfected as in B and the relative luciferase activity was determined. (**D**) *Atf6^-/-^* and *Atf6/Xbp1* dko cells were transfected as in B and the relative luciferase activity was determined. (**E**) *Xbp1^-/-^* MEFs were co-transfected with firefly and renilla luciferase vectors as in B. Expression vectors for XBP1s, XBP1u, ATF6(N), or empty vector (EV) were co-transfected. Relative luciferase activities (firefly: renilla) ± SEM of 3 biological replicates are shown. (**F**) *Atf6^-/-^* cells were transfected as in E and the relative luciferase activity was determined. (**G**) *Atf6/Xbp1* dko cells were transfected as in E and the relative luciferase activity was determined. Data provided in *Figure 7—source data 1*.

The online version of this article includes the following source data for figure 7:

**Source data 1.** Data points of luciferase assays.

suggesting that motif 4 is sufficient to confer MIEP responsiveness to XBP1s and ATF6. To determine whether motif 4 is also of crucial importance for MCMV replication, we mutated motif four within the MCMV genome. The resulting virus, MCMV-GFP-4-mut, was compared to the parental MCMV-GFP virus in a multi-step replication kinetic analysis (*Figure 9C*). Indeed, viral replication was massively reduced when motif 4 within the MIEP was mutated. This result confirmed that the XBP1 and

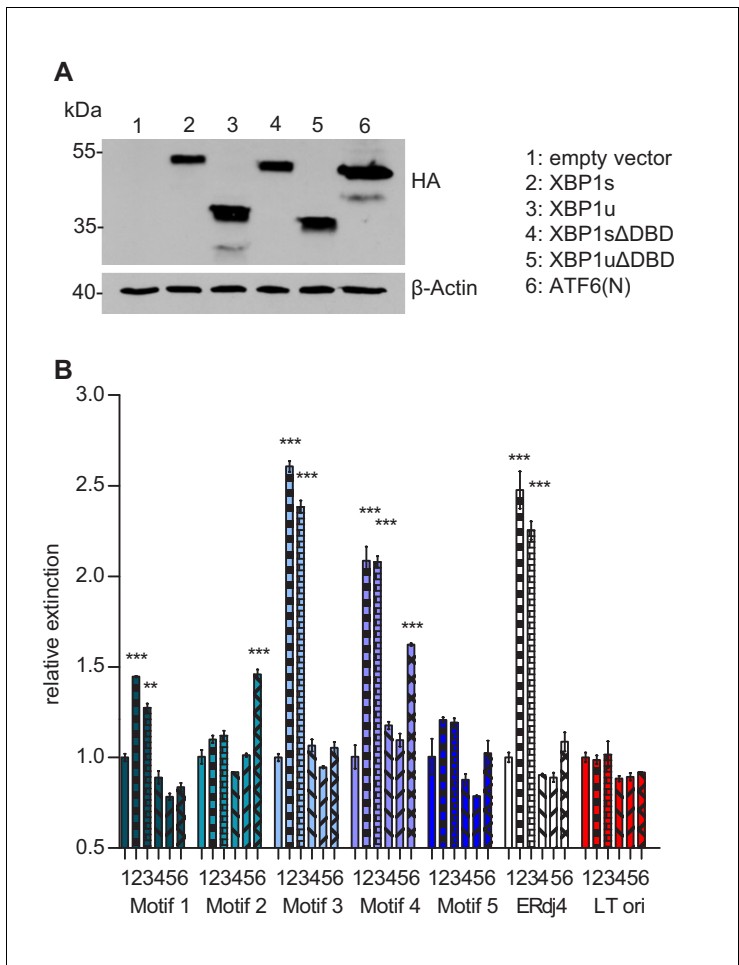

**Figure 8.** Transcription factor binding the MIE promotor. (**A**) HEK 293A cells were transfected with expression vectors encoding HA-tagged WT or mutant XBP1 and ATF6(N) transcription factors. Nuclear extracts were obtained, and transcription factor expression was verified by immunoblot analysis (representative of 2 independent experiments). (**B**) Microtiter plates coated with dsDNA oligonucleotides containing XBP1 core binding motifs from the MCMV major immediate-early promoter, the *ERdj4* promoter (positive control) or an unrelated sequence from the SV40 origin of replication (negative control). Wells were incubated with nuclear extracts 1 to 6 shown in A, and transcription factor binding was measured by DPI-ELISA, and values were normalized to extract 1 (empty vector). Means ± SEM of 3 biological replicates are shown. Significance was determined for all values above the cut-off (1.25). **, p<0.01; ***, p<0.001. Data provided in *Figure 8—source data 1*.

The online version of this article includes the following source data for figure 8:

**Source data 1.** Data points of the DPI-ELISA.

ATF6(N) target motif identified in DNA-binding and luciferase reporter assays is required for efficient replication of the virus.

Taken together, the results of this study show that MCMV transiently activates the IRE1-XBP1 signaling pathway in the early phase of infection in order to relieve XBP1u-mediated repression of viral gene expression and replication. The MCMV major immediate-early promoter requires XBP1s or ATF6(N) binding to motif 4 for full activation, and XBP1u prevents promoter activity by both TFs. Thus, XBP1u acts as a key repressor of two UPR pathways, the IRE1-XBP1 and the ATF6 pathways, which have overlapping and redundant functions.

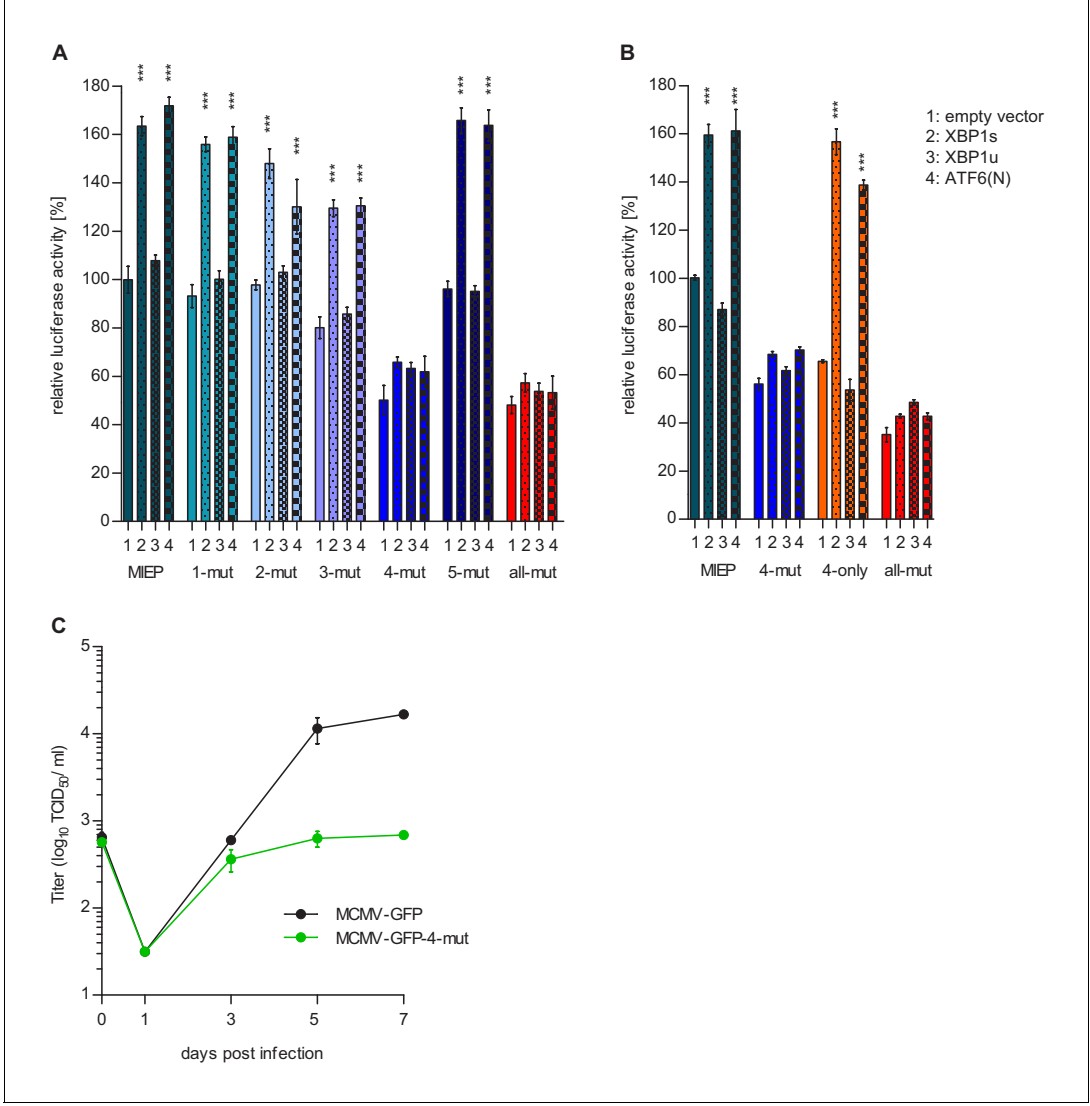

**Figure 9.** Motif 4 is necessary and sufficient for MIEP activation by XBP1s and ATF6(N). (**A**) *Atf6/Xbp1* dko MEFs cells were transfected with a firefly luciferase vector containing either the WT major immediate-early promotor (MIEP) or a MIEP with one or all ACGT motifs mutated. Expression vectors for XBP1s, XBP1u, ATF6(N), or empty vector (EV) were co-transfected. Renilla luciferase was expressed by co-transfection and used for normalization. Relative luciferase activities (firefly: renilla) ± SEM of 3 biological replicates are shown. ***, p<0.001; all other differences were not significant (p>0.05). (**B**) *Atf6/Xbp1* dko MEF cells were transfected and analyzed as in A. In addition, a MIEP vector with all ACGT motifs mutated except motif 4 (4-only) was included. (**C**) Multistep MCMV replication kinetics in WT MEF cells. Cells were infected with MCMV-GFP or MCMV-GFP-4-mut (MOI 0.1), respectively. Virus titers in the supernatants were determined by titration and are shown as means ± SEM of 3 biological replicates. Data provided in *Figure 9—source data 1*.

The online version of this article includes the following source data for figure 9:

**Source data 1.** Data points of luciferase assays and growth curve.

## Discussion

This study shows how MCMV harnesses UPR signaling to regulate its own life cycle. The viral MIEP, the key promotor of the lytic replication cycle, contains five ACGT motifs, of which motif 4 is most important for MIEP activation by XBP1s or ATF6(N) (*Figure 9*). The fact that MIEP activity and viral replication are virtually unaffected in the absence of either XBP1 or ATF6 but are massively reduced in the absence of both, strongly suggests that the two UPR TFs have redundant roles in the activation of the MIEP. The most important finding, however, is the dominant role of XBP1u as a repressor of viral gene expression and replication. When IRE1 is absent or its RNase activity is blocked, XBP1u

dominates and prevents MIEP activation by XBP1s and ATF6(N). Thus, the transient activation of IRE1 during the early phase of MCMV infection serves the virus by increasing XBP1s and decreasing XBP1u expression, thereby relieving XPB1u-mediated repression. At present, the MCMV protein(s) responsible for the early activation of IRE1 remains unknown. Most of the viral envelope glycoproteins are expressed at late times post infection (i.e., starting 12 hpi) and are thus unlikely to be responsible. However, it is conceivable that certain early glycoproteins are particularly potent stress inducers, either by binding and sequestering BiP (*Bakunts et al., 2017*; *Vitale et al., 2019*) or by inducing ER calcium release. Possible candidates for the latter include the MCMV homolog of the HCMV UL37x1 protein (*Sharon-Friling et al., 2006*) or the viral G protein-coupled receptors, which can activate phospholipase C and the inositol trisphosphate receptor (*Boeck and Spencer, 2017*; *Sherrill et al., 2009*).

How exactly XBP1u antagonizes XBP1s and ATF6(N)-mediated promoter activation, remains to be determined. Two possible mechanisms may be involved: XBP1u could interact with XBP1s and ATF6(N) and target them for proteasomal degradation as previously reported (*Yoshida et al., 2006*; *Yoshida et al., 2009*). Additionally, XBP1u could bind directly to DNA and inhibit promoter activation by XBP1s and ATF6(N). Indeed, we show that XBP1u is fully capable of binding to the same DNA sequences that XBP1s binds to (*Figure 8*). Moreover, a truncated XBP1 protein, XBP1stop, which lacks the C-terminal destabilizing domain of XBP1u, had a similar repressive effect as XBP1u (*Figure 5D*) indicating that destabilization is not required for repression.

Ever since its discovery, the importance of XBP1u has been the subject of controversy. In yeast, the unspliced mRNA of HAC1, the yeast homolog of XBP1, is posttranscriptionally silenced, and a protein product of the unspliced HAC1 mRNA has not been detected (*Rüegsegger et al., 2001*). In mammalian cells, the XBP1u protein is readily detectable, but XBP1u has a short half-life (*Navon et al., 2010*; *Tirosh et al., 2006*). This has led to the conclusion that XBP1u is of minor importance and its function restricted to fine-tuning of the UPR (*Byrd and Brewer, 2012*). Hence, many review articles on the UPR barely mention XBP1u. On the other hand, XBP1u has been shown to dimerize with XBP1s and ATF6(N) and destabilize them (*Yoshida et al., 2006*; *Yoshida et al., 2009*), suggesting a more important regulatory role. However, the physiological relevance of these findings has been called into question by some scientists because the findings were made in cells overexpressing XBP1u. The results of our study show that XBP1u plays a very important role as a repressor of XBP1s and ATF6(N)-mediated activation of the MCMV MIEP, which results in a massively (up to 100-fold) reduced production of progeny virus (*Figures 2*, *3B* and *5D*). In IRE1-deficient cells, the *Xbp1* transcript is expressed from its endogenous promoter, not from a strong heterologous promoter. Thus, the observed effects cannot be dismissed as overexpression artifacts. Moreover, the fact that MCMV replication is massively reduced in *Ern1* ko cells (*Figures 2* and *3B*), but not in *Ern1/Xbp1* dko cells (*Figure 5B*), demonstrates that XBP1u expression rather than the absence of RIDD is responsible for the observed effect. Hence, the data of our study suggest that XBP1u plays an unexpectedly important role as a repressor, at least under the conditions of viral infection. Whether XBP1u can repress expression of cellular genes in a similar fashion is an important question that needs to be answered in future studies.

XBP1 binding sites have also been identified in immediate-early gene promoters of the γ-herpesviruses Epstein-Barr virus, Kaposi's sarcoma-associated herpesvirus, and murine gammaherpesvirus 68 (MHV-68), suggesting a potential role for XBP1s in transactivating these promoters during reactivation from latency (*Bhende et al., 2007*; *Matar et al., 2014*; *Sun and Thorley-Lawson, 2007*; *Wilson et al., 2007*; *Yu et al., 2007*). However, one study demonstrated that XBP1 was not required for MHV-68 reactivation in B cells (*Matar et al., 2014*). The authors of the study speculated that the apparent independence of MHV-68 reactivation from XBP1 expression in B cells might reflect redundancy among CREB/ATF family TFs. In light of our data showing a redundancy of XBP1s and ATF6(N) in the activation of the MCMV MIEP, this conjecture may well be correct. Conversely, it would be interesting to know whether XBP1 and ATF6 also regulate MCMV reactivation from latency. Unfortunately, this obvious question is difficult to address as there is no manageable cell culture system for MCMV latency and reactivation. Nonetheless, it appears likely that β and γ-herpesviruses harness UPR TFs in a similar fashion to regulate their life cycle.

# Materials and methods

## Key resources table

| Reagent type (species) or resource | Designation | Source or reference | Identifiers | Additional information |
|---|---|---|---|---|
| Strain, strain background (*E. coli*) | GS1783 | PMID:16526409 | | BAC host for recombineering |
| Strain, strain background (*murid herpesvirus 1*) | MCMV-GFP | PMID:11209080 | | BAC-derived MCMV (Smith strain) expressing GFP |
| Cell line (*H. sapiens*) | HEK 293A | Invitrogen | R705-07; RRID:CVCL_6910 | Transformed human embryonic kidney cells |
| Cell line (*H. sapiens*) | HEK 293T | ATCC | CL-11268 | |
| Cell line (*H. sapiens*) | Phoenix | ATCC | CRL-3213; RRID:SCR_003163 | Packaging cell line for retrovirus production |
| Cell line (*M. musculus*) | WT MEF | PMID:19364921 | | SV40 TAg immortalized MEF |
| Cell line (*M. musculus*) | 10.1 | PMID:1752433 | | spontaneously immortalized BALB/c MEF |
| Cell line (*M. musculus*) | *Ern1*$^{-/-}$ MEF | PMID:11780124 | | IRE1-deficient MEF |
| Cell line (*M. musculus*) | TetON-*Ern1*$^{-/-}$ | This paper | | *Ern1*$^{-/-}$ MEFs expressing rtTA (TetON) |
| Cell line (*M. musculus*) | TetON-IRE1-GFP | This paper | | TetON-*Ern1*$^{-/-}$ cells, inducible expression of IRE1-GFP |
| Cell line (*M. musculus*) | *Xbp1*$^{-/-}$ MEF | PMID:10652269 | | |
| Cell line (*M. musculus*) | *Ern1* ko MEF | This paper | | WT MEF, *Ern1* knocked out by CRISPR/Cas9 |
| Cell line (*M. musculus*) | *Xbp1* ko MEF | This paper | | WT MEF, *Xbp1* knocked out by CRISPR/Cas9 |
| Cell line (*M. musculus*) | *Traf2* ko MEF | This paper | | WT MEF, *Traf2* knocked out by CRISPR/Cas9 |
| Cell line (*M. musculus*) | *Atf6*$^{-/-}$ MEF (LT) | This paper PMID:17765679 | | SV40 TAg immortalized MEF |
| Cell line (*M. musculus*) | *Atf6*$^{+/+}$ MEF (LT) | This paper PMID:17765679 | | SV40 TAg immortalized MEF |
| Cell line (*M. musculus*) | *Atf6/Xbp1* dko | This paper | | *Atf6*$^{-/-}$ MEF, *Xbp1* knocked out by CRISPR/Cas9 |
| Recombinant DNA reagent | pMSCVhygro, pMSCVpuro | Clontech | | Retroviral vector plasmids |
| Recombinant DNA reagent | pGL3-Basic | Promega | E1751 | Firefly luciferase reporter plasmid |
| Recombinant DNA reagent | pGL4.73 | Promega | E6911 | Renilla luciferase control plasmid |
| Antibody | anti-HA (mouse monoclonal) | Covance | MMS-101P; RRID:AB_291259 | WB (1:1000) |
| Antibody | anti-IRE1 (rabbit monoclonal) | Cell Signaling | Cat# 3294; RRID:AB_823545 | WB (1:1000) |
| Antibody | anti-pIRE1 (rabbit polyclonal) | Novus Biologicals | NB100-2323; RRID:AB_10145203 | WB (1:500) |

*Continued on next page*

*Continued*

| Reagent type (species) or resource | Designation | Source or reference | Identifiers | Additional information |
|---|---|---|---|---|
| Antibody | anti-XBP1 (rabbit polyclonal) | Santa Cruz | sc-7160; RRID:AB_794171 | WB (1:500) |
| Antibody | anti-TRAF2 (rabbit monoclonal) | Cell Signaling | Cat# 4724; RRID:AB_2209845 | WB (1:1000) |
| Antibody | anti-IE1 (Chroma101, mouse) | Stipan Jonjic (Univ.of Rijeka) | | WB (1:1000) |
| Antibody | anti-M57 (mouse) | Stipan Jonjic (Univ.of Rijeka) | | WB (1:1000) |
| Antibody | anti-gB (M55.01, mouse) | Stipan Jonjic (Univ.of Rijeka) | | WB (1:1000) |
| Antibody | anti-GFP (mouse monoclonal) | Roche | Cat# 11814460001; RRID:AB_390913 | WB (1:1000) |
| Antibody | anti-β-Actin (mouse monoclonal) | Sigma-Aldrich | A2228; RRID:AB_476697 | WB (1:1000) |
| Chemical compound, drug | GenJet | SignaGen Laboratories | SL100489-MEF | Transfection reagent |
| Chemical compound, drug | Polyethylenimine (PEI) | Sigma-Aldrich | 764604 | Transfection reagent |
| Commercial assay or kit | CellTiter-Glo Luminescent Cell Viability Assay | Promega | G7570 | |
| Commercial assay or kit | Dual-Luciferase Reporter Assay System | Promega | E1910 | |

## Cells and viruses

The following immortalized fibroblast lines were used: wildtype MEFs (*Manzl et al., 2009*), 10.1 cells (*Harvey and Levine, 1991*), *Ern1*$^{-/-}$ MEFs (*Calfon et al., 2002*), and *Xbp1*$^{-/-}$ MEFs (*Reimold et al., 2000*). Primary *Atf6*$^{+/+}$ and *Atf6*$^{-/-}$ MEFs (*Wu et al., 2007*) were kindly provided by D. Thomas Rutkowski (University of Iowa, USA). They were immortalized by transduction with a retroviral vector encoding SV40 large T antigen (pBABE-zeo largeTcDNA, Addgene). The *Atf6*$^{+/+}$ vs. *Atf6*$^{-/-}$ state was verified by PCR as described (*Wu et al., 2007*). TetON-MEFs were obtained by transduction of wiltype MEFs with a lentiviral vector encoding the TetON transactivator (pLIX-402, Addgene). TetON-*Ern1*$^{-/-}$ cells were generated in a similar fashion. TetON-IRE1-GFP MEFs were obtained by reconstituting TetON-*Ern1*$^{-/-}$ MEFs with GFP-tagged murine IRE1 under control of a 'tight' Tet-responsive element (Clontech) essentially as described (*Bakunts et al., 2017*; *Cohen et al., 2017*). The GFP tag was introduced into the juxtamembrane cytosolic linker domain of IRE1, where such tagging had been shown before to not interfere with function of human IRE1 (*Li et al., 2010*). All cells were grown under standard conditions in Dulbecco´s modified Eagle´s medium supplemented with 10% fetal calf serum or 10% fetal calf serum tetracycline-free, 100 IU/ml penicillin and 100 µg/ml streptomycin (Sigma). All cells were tested regularly for mycoplasma contamination and were found to be negative.

MCMV-GFP (*Brune et al., 2001*) was propagated and titrated on 10.1 fibroblasts. Viral titers were determined by using the median tissue culture infective dose (TCID$_{50}$) method. Virus was inactivated by 254-nm-wavelength UV irradiation with 1 J/cm$^2$ for 30 s using a UV cross-linker (HL-2000 HybriLinker; UVP).

## Mutagenesis of the MCMV MIEP

The recombinant MCMV carrying an ACGT-to-CTAG mutation of MIEP motif 4 was constructed by BAC (bacterial artificial chromosome) recombineering using the *en passant* mutagenesis protocol (*Tischer et al., 2010*). A kanamycin resistant gene was PCR-amplified using primers MIEP_4_mut fwd (5'-GGTACTTTCCCATAGCTGATTAATGGGAAAGTACCGTTCTCGAGCCAATACCTAGCAA TGGGAAGTGAAAGGGCAGtagggataacagggtaatcgattt-3') and MIEP_4-mut rev (5'-

GGGGAAAACCGGGGCGGTGTTACGTTTTGGCTGCCCTTTCACTTCCCATTGCTAGGTATTGGC
TCGAGAACGGTACTgccagtgttacaaccaattaacc-3'). The linear PCR product was used for homologous recombination in *Escherichia coli* strain GS1783 containing the MCMV-GFP BAC. The kanamycin resistance cassette was used for positive selection in the first recombination step and removed in the second step (*Tischer et al., 2010*). The MCMV-GFP-4-mut BAC was examined by restriction fragment analysis and sequencing of the mutated region. Mutant and control viruses were reconstituted by BAC transfection of mouse fibroblast using GenJet transfection reagent (SignaGen).

## Plasmids and transfection

XBP1-wt and XBP1-unsplicable cDNAs (*Tirosh et al., 2006*) were cloned in pcDNA3. The XBP1-wt vector was used to generate by PCR an XBP1stop mutant consisting of only the N-terminal domain of XBP1. The XBP1s cDNA was generated by isolating RNA from MEFs stimulated with tunicamycin (Sigma), reverse transcription, and PCR amplification. To generate XBP1-unsplicable and XBP1-spliced plasmids lacking the DNA-binding domain (ΔDBD), the complete DBD was replaced by an alternative NLS sequence as previously described (*Zhou et al., 2011*). Transcripts of XBP1 proteins were HA-tagged by PCR amplification and subcloned in pMSCVpuro (Clonetech), a retroviral vector plasmid. Similarly, mIRE1-wt was PCR-amplified (without a myc tag) from pcDNA3-mIRE1-3xmyc (*Stahl et al., 2013*) and subcloned in pMSCVhygro (Clontech) using BglII and HpaI sites. The K907A mutation was introduced by QuikChange site-directed mutagenesis (Stratagene). Transient transfection was done using GenJet (SignaGen) or polyethylenimine (Sigma).

## Retroviral transduction and CRISPR/Cas9 gene editing

Retrovirus production using the Phoenix packaging cell line and transduction of target cells was done as described (*Swift et al., 2001*). Cells transduced with MSCVpuro and MSCVhygro vectors were selected with 1.5 µg/ml puromycin (Sigma) or 50 µg/ml hygromycin B (Roth), respectively.

Guide RNAs (*Supplementary file 3*) for genes of interest were designed using the online tool E-CRISP (http://www.e-crisp.org/E-CRISP/designcrispr.html) and inserted into the lentiviral vector pSicoR-CRISPR-puroR (kindly provided by R. J. Lebbink, University Medical Center Utrecht, Netherlands). Lentiviruses were produced in 293 T cells using standard third-generation packaging vectors as described (*van Diemen et al., 2016*). Lentiviruses were used to transduce MEFs in the presence of polybrene (5 µg/ml). Cells were selected with 1.5 µg/ml puromycin and single cell clones were obtained by limiting dilution.

## Immunoblot analysis

Whole cell lysates were obtained by lysing cells in RIPA buffer supplemented with a cOmplete, Mini protease inhibitor cocktail (Roche). XBP1s, XBP1u, and ATF6(N) were extracted from cells treated with thapsigargin (Sigma) using a nuclear extraction protocol (*Stahl et al., 2013*). Insoluble material was removed by centrifugation. Protein concentrations were measured using a BCA assay (Thermo Fisher Scientific). Equal protein amounts were boiled in sample buffer and subjected to SDS-PAGE and semi-dry blotting onto nitrocellulose membranes. For immunodetection, antibodies against the following epitopes were used: HA (16B12, Covance), β-actin (AC-74, Sigma), XBP1 (M-186, Santa Cruz), IRE1 (14C10, Cell Signaling), pIRE1 (NB100-2323, Novus Biologicals), TRAF2 (Cell Signaling), and GFP (Roche). Antibodies against MCMV IE1 (CROMA101), M57, and M55/gB (SN1.07) were provided by Stipan Jonjic (University of Rijeka, Croatia). Secondary antibodies coupled to horseradish peroxidase (HRP) were purchased from Dako.

## RNA isolation and quantitative PCR

Total RNA was isolated from MEFs using an innuPREP RNA Mini Kit (Analytik-Jena). Reverse transcription and cDNA synthesis was carried out with 2 µg RNA using 200 U RevertAid H Minus Reverse Transcriptase, 100 pmol Oligo(dT)$_{18}$, and 20 U RNase inhibitor (Thermo Fisher Scientific). Quantitative real-time PCR reactions employing SYBR Green were run in a 7900HT Fast Real-Time PCR System (Applied Biosystems). The following primers were used: 5'-GAGTCCGCAGCAGGTG-3' and 5'-GTGTCAGAGTCCATGGGA-3' for *Xbp1s*, 5'-GTGTCAGAGTCCATGGGA-3' and 5'-GTGTCAGAGTCCATGGGA-3' for *Xbp1u,* and 5'-CCCACTCTTCCACCTTCGATG-3' and 5'-GTCCACCACCCTGTTGCTGTAG-3' for *Gapdh*. Primers for the amplification of IE1 (M123), E1 (M112), gB (M55), and

M37 transcripts have been described previously (*Chapa et al., 2013*; *Chapa et al., 2014*). Reactions were performed under the following conditions: 45 cycles of 3 s at 95°C and 30 s at 60°C. Three replicates were analyzed for each condition, and the relative amounts of mRNAs were calculated from the comparative threshold cycle (Ct) values by using *Gapdh* as reference.

### Replication kinetics

Cells were seeded in 6-well plates and infected by an MOI of 3 or 0.1 for single or multi-step replication kinetics, respectively. Six hpi the medium was exchanged to remove the inoculum. Supernatant samples were harvested at different times post infection, and viral titers were determined on 10.1 fibroblasts using the $TCID_{50}$ method.

### MIE promotor activity assay

The firefly luciferase reporter vector pGL3-Basic, the renilla luciferase control vector pGL4.73, and the Dual-Glo Luciferase assay system were purchased from Promega. WT and mutant versions of the MCMV MIE promotor (*Supplementary file 1*) were synthesized by Integrated DNA Technologies and cloned in pGL3-Basic. Cells were transfect in 6-well dishes using GenJet with pGL3-MIEP (0.5 µg), pGL4.73 (0.05 µg) and transcription factor expression plasmids. The total amount of DNA was kept constant at 3 µg by filling up with empty pcDNA3 vector. On the following day, the medium was exchanged and cells were incubated for another 24 hr. Cells were harvested in lysis buffer and luciferase activity was measured using a Dual-Glo Luciferase assay and a luminescence plate reader (FLUOstar-Omega, BMG Labtech). Firefly and Renilla luciferase activities were evaluated for each sample. At least three biological replicates were used for each condition.

### DNA-Protein interaction ELISA (DPI-ELISA)

The DPI-ELISA was performed essentially as described in detail elsewhere (*Brand et al., 2010*; *Underwood et al., 2013*). Terminally biotinylated dsDNA oligonucleotides containing 3 copies of putative TF binding sites (*Supplementary file 2*) were purchased from Eurofins. Oligonucleotides were adsorbed to streptavidin-coated 96-well microtiter plates and incubated with nuclear extracts (10 µg protein) from transfected HEK 293A cells expressing the HA-tagged XBP1 or ATF6(N) proteins. Nuclear extracts were obtained as described (*Stahl et al., 2013*). TF binding to DNA was quantified by ELISA using an anti-HA antibody, an HRP-coupled secondary antibody, and an ABTS substrate (Roche). Absorbance at 405 nm wavelength was measured using a FLUOstar-Omega plate reader. At least three biological replicates were used for each condition.

### Cell viability assay

TetON-IRE1-GFP cells were seeded in 96-well plates at a density of 1000 cells per well. Cells were treated with 0, 10 or 25 nM doxycycline at the time of seeding. Medium was replenished 1 and 3 days after seeding. Cell viability was determined on day 3 and day 5 by measuring intracellular ATP levels with a Cell Titer-Glo Luminescent Cell Viability Assay kit (Promega) and a FLUOstar Omega luminometer (BMG Labtech). The values were normalized to non-stimulated cells.

### Statistical analysis

All statistical analyses were performed with GraphPad Prism 5.0 software. One-way ANOVA followed by Bonferroni´s post hoc test was used for the analysis of the luciferase reporter assays, the cell viability assay, and the DPI-ELISA.

## Acknowledgements

We thank D Thomas Rutkowski for providing *Atf6*$^{-/-}$ MEFs and Svenja Siebels for helpful advice on the DPI-ELISA. This study was supported by the Deutsche Forschungsgemeinschaft (BR 1730/6-1 to WB), the EU Horizon 2020 Marie Sklodowska-Curie ITN-TREATMENT (Grant 721236 to BT), the German-Israeli Foundation (Grant I-1471–414.13/2018 to BT), and Giovanni Armenise-Harvard Foundation (Career Development Award to EvA).

## Additional information

### Competing interests

Eelco van Anken: Reviewing editor, *eLife*. The other authors declare that no competing interests exist.

### Funding

| Funder | Grant reference number | Author |
|---|---|---|
| Deutsche Forschungsgemeinschaft | BR1730/6-1 | Wolfram Brune |
| H2020 Marie Skłodowska-Curie Actions | ITN-TREATMENT grant 721236 | Boaz Tirosh |
| German-Israeli Foundation for Scientific Research and Development | Grant I-1471- 414.13/2018 | Boaz Tirosh |
| Giovanni Armenise-Harvard Foundation | Career Development Award | Eelco van Anken |

The funders had no role in study design, data collection and interpretation, or the decision to submit the work for publication.

### Author contributions

Florian Hinte, Conceptualization, Investigation, Methodology, Writing - original draft, Writing - review and editing; Eelco van Anken, Resources, Methodology, Writing - review and editing; Boaz Tirosh, Conceptualization, Resources, Writing - review and editing; Wolfram Brune, Conceptualization, Supervision, Funding acquisition, Writing - original draft, Writing - review and editing

### Author ORCIDs

Florian Hinte  https://orcid.org/0000-0003-0820-7041
Eelco van Anken  http://orcid.org/0000-0001-9529-2701
Boaz Tirosh  http://orcid.org/0000-0001-8067-6577
Wolfram Brune  https://orcid.org/0000-0002-6078-5255

### Decision letter and Author response

Decision letter https://doi.org/10.7554/eLife.51804.sa1
Author response https://doi.org/10.7554/eLife.51804.sa2

## Additional files

### Supplementary files

• Supplementary file 1. Wildtyp and mutated Major Immediate-Early Promoter (MIEP) sequences. Minimal XBP1 binding motifs (ACGT) are underlined (black), mutated motifs are in red. The TATA box is in bold.

• Supplementary file 2. Oligonucleotides used for the DNA-Protein Interaction (DPI) ELISA.

• Supplementary file 3. Guide RNAs (gRNAs) used for CRISPR/Cas9 gene editing.

• Transparent reporting form

### Data availability

All data generated or analysed during this study are included in the manuscript and supporting files. Source data files have been provided for Figures 1 through 9.

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
