## [Decision Letter]

**Acceptance summary:**

This study is focused on an understudied aspect of virus-host interactions, the effects of UPR pathways on lytic herpesvirus replication. Importantly, very little is known about XBP1u effects on any virus, and the discovery of a repressive function for XBP1u provides a tidy explanation for IRE1 activation observed during early lytic replication. The study makes good use of CRISPR technology to knock out various components of the UPR pathway to identify genetic determinants of these responses. It is a well-investigated example of how a virus can both take advantage of a cellular pathway and subsequently repress it during a single infectious cycle.

**Decision letter after peer review:**

Thank you for submitting your article "Repression of viral gene expression and replication by the unfolded protein response effector XBP1u" for consideration by *eLife*. Your article has been reviewed by three peer reviewers, and the evaluation has been overseen by Karla Kirkegaard as the Senior and Reviewing Editor. The following individuals involved in review of your submission have agreed to reveal their identity: Hiderou Yoshida (Reviewer #2); Craig McCormick (Reviewer #3).

The reviewers have discussed the reviews with one another and the Reviewing Editor has drafted this decision to help you prepare a revised submission.

Summary:

The endoplasmic reticulum (ER) is an organelle where secretory and membrane proteins are synthesized and folded with the assistance of ER chaperones. When synthesis of secretory or membrane proteins is increased and overwhelms the capacity of the ER, unfolded proteins are accumulated in the ER (ER stress), resulting in ER stress-induced cell death. To cope with ER stress, eukaryotic cells activates a cytoprotective mechanism called the ER stress response (also called the unfolded protein response (UPR)) in order to augment the capacity of the ER. In mammalian cells, the UPR consists of three response pathways, that is, the ATF6, IRE1 and PERK pathways. Each pathway is regulated by a specific sensor and a transcription factor. For instance, pATF6(P) and pATF6(N) are a sensor and a transcription factor for the ATF6 pathway, while IRE1 and pXBP1(S) are those for the IRE1 pathway. Mammalian XBP1 gene encodes two proteins, that is, pXBP1(U) and pXBP1(S). In normal growth conditions, pXBP1(U) protein is translated from unspliced XBP1 mRNA, whereas upon ER stress, activated IRE1 converts unspliced XBP1 mRNA into spliced mRNA, from which pXBP1(S) is translated. pXBP1(S) is an active transcription factor that induces transcription of ER-related genes, while pXBP1(U) has several functions, including transcriptional suppression against pXBP1(S) and pATF6(N), and anchoring unspliced mRNA to ER membrane.

During viral replication, a number of viral proteins are synthesized and activate the UPR to support the synthesis of their proteins. On the other hand, the system of the ER-associated degradation (ERAD) augmented by the IRE1 pathway degrades viral proteins and the PERK pathway attenuates translation of viral proteins. Thus, the UPR is beneficial as well as harmful to virus, and in some cases virus modulate the UPR of host cells to ensure propagation of their offspring.

It has been revealed that cytomegalovirus (CMV) modulates the mammalian UPR. It limits translational repression of their proteins by the PERK pathway, and enhances folding of their proteins through induced expression of ER chaperones by the ATF6 pathway. Interestingly, the authors previously revealed that the UPR is activated during the early phase of viral replication, while CMV reduces expression of IRE1 by their proteins (M50 and UL50) during the late phase of the replication.

In this manuscript, the authors examined the mechanism and the biological significance of the transient activation of the UPR in the early phase of CMV replication. They found that moderate activation of RNase activity of IRE1 is beneficial to replication. Interestingly, they showed that pXBP1(U) can suppress replication, while pATF6(N) and pXBP1(S) are beneficial for replication by upregulating transcription of the viral major immediate-early promoter (MIEP) genes through the Motif 4 enhancer element. From these observations, the authors concluded that CMV harnesses the UPR to regulate its own life cycle by activating the IRE1-XBP1 pathway during the early post infection to relieve repression by pXBP1(U). Their findings are important not only for viral research but also for the research of the UPR, since they revealed the redundant function of pXBP1(S) and pATF6(N) as activators of the viral life cycle, and an unexpected role of pXBP1(U) as a potent repressor of both pXBP1(S) and pATF6(N)-mediated activation.

That the product of an unspliced version of a cellular message is subverted by a virus is extremely interesting, and this work is well-executed. However, the timing of this is obviously of great interest, and the reviewers' major requests were explications of that timing for all the relevant players.

Essential revisions:

1) All reviewers had questions about the timing of the experiments. Examples of individual reviewer comments to this effect are below. What is needed is no less than the abundance of each of the relevant molecules as a function of time throughout the viral infection. At least, filling out the tests at the time points requested by the reviewers below is needed to clarify all the moving parts in this complex system.

a) The main comment addresses the kinetics of the effects of IRE1 activation. Figure 1 shows XBP1u mRNA gets spliced at 6-7 hours after infection with a rapid decline of XBP1u splicing from 8-24 hours. However, all subsequent experiments on the effects of XBP1u were examined between 1 and 9 days after infection. Thus, it is unclear whether early effects were studied.

b) Figure 2. It is puzzling that viral replication is the same in the absence (0nM dox) or high abundance (25nM dox) of IRE1. In contrast, intermediate abundances of IRE1 stimulate viral replication. Is cell viability, and consequently viral replication, affected at 25nM dox? Alternatively, are late effects, i.e. inhibition of IRE1, already monitored after 5 days? The late glycoprotein B is already observed at 2 days after infection (Figure 3D-F).

c) Figure 3. The effect of IRE1 ko on viral replication is very clear; IRE1 deficiency drops multiround replication by several logs. The diminished viral protein accumulation on the western blots in Figure 3D is quite subtle by comparison, but understandable because it is a high MOI single-cycle infection rather than low MOI multiround. Essentially, it is difficult to draw a strong conclusion from these western blots, and they certainly could use more support in the form of RT-qPCR for viral transcripts (which would not be limited to available antibodies, and could be used in concert with multi-round infection, as in Figure 3B.)

d) Some conclusions are not adequately supported by the data. For example, while the authors provide evidence for transient IRE1 activation during MCMV infection via an RT-qPCR assay that measures XBP1 mRNA splicing, they do not corroborate this data with more direct evidence of IRE1 activation via immunoblotting for IRE1 phosphorylation or XBP1s protein accumulation, which is standard practice in the UPR field. Moreover, they do not investigate whether MCMV infection (in either low MOI or high MOI infections) affects the levels or activation of IRE1, XBP1, or any other UPR component. In this regard, application of the RT-qPCR assay for XBP1 splicing to an MCMV infection time course would be quite informative.

e) It remains unclear whether MCMV infection causes alterations in the levels or activation of any of the cellular UPR machinery like IRE1, XBP1s or ATF6. At the least, this RT-qPCR data should be accompanied by western blots that show clear accumulation of XBP1s protein, so the authors can confidently conclude that IRE1 is indeed being activated and having the anticipated effect on XBP1s protein accumulation. The authors should directly demonstrate IRE1 activation in response to infection using conventional western blotting techniques, looking for IRE1 phosphorylation and impeded electrophoretic mobility, or accumulation of XBP1u or XBP1s proteins.

2) In Figure 2, it remains formally possible that higher doses of dox in Figure 2 negatively impact MCMV replication in an IRE1-independent manner. Importantly, this experiment lacks an empty vector control that would clarify this matter.

3) The data are remarkably consistent between biological replicates. Are these true biological replicates performed at different times, or cells infected in parallel with the same virus stocks on the same day? Some clarity on this point, and a look at the primary data, would be illuminating.

4) This study would be significantly strengthened by identification of a viral factor responsible for IRE1 activation; this may be beyond the scope of the current study, but the reader would benefit from understanding approaches to identify such a factor among MCMV early gene products.

---

## [Author Response]

Essential revisions:1) All reviewers had questions about the timing of the experiments. Examples of individual reviewer comments to this effect are below. What is needed is no less than the abundance of each of the relevant molecules as a function of time throughout the viral infection. At least, filling out the tests at the time points requested by the reviewers below is needed to clarify all the moving parts in this complex system.a) The main comment addresses the kinetics of the effects of IRE1 activation. Figure 1 shows XBP1u mRNA gets spliced at 6-7 hours after infection with a rapid decline of XBP1u splicing from 8-24 hours. However, all subsequent experiments on the effects of XBP1u were examined between 1 and 9 days after infection. Thus, it is unclear whether early effects were studied.

Yes, early effects on viral protein expression (after high-MOI infection) were analyzed by immunoblot and are shown in Figure 3D. Moreover, Marcinowski et al., 2012 have reported an upregulation of cellular ER stress response genes at 5-6 hours post infection. Additionally, we have extended Figure 1 by adding an immunoblot detection of pIRE1 and XBP1s to corroborate the qRT-PCR data (see response to comments 1d and e).

b) Figure 2. It is puzzling that viral replication is the same in the absence (0nM dox) or high abundance (25nM dox) of IRE1. In contrast, intermediate abundances of IRE1 stimulate viral replication. Is cell viability, and consequently viral replication, affected at 25nM dox? Alternatively, are late effects, i.e. inhibition of IRE1, already monitored after 5 days? The late glycoprotein B is already observed at 2 days after infection (Figure 3D-F).

The reviewers ask why strong induction of IRE1 expression with 25 nM doxycycline is detrimental for viral replication and suggest that cell viability might be compromised. We fully agree that this is the most likely explanation. A seminal paper by Feroz Papa’s laboratory has shown that strong induction of IRE1 expression leads to its autoactivation, ER stress, and apoptosis (Han et al., 2009). We have measured the viability of the IRE1-GFP expressing fibroblasts on day 3 and 5 after induction with different concentrations of doxycycline and found a significantly decreased viability upon induction with 25 nM dox. In the revised manuscript we state that impaired cell viability is probably the reason for the low virus titers obtained after induction with high dox concentrations and refer to the paper of Han et al. as well as our own data (included as Figure 2D).

c) Figure 3. The effect of IRE1 ko on viral replication is very clear; IRE1 deficiency drops multiround replication by several logs. The diminished viral protein accumulation on the western blots in Figure 3D is quite subtle by comparison, but understandable because it is a high MOI single-cycle infection rather than low MOI multiround. Essentially, it is difficult to draw a strong conclusion from these western blots, and they certainly could use more support in the form of RT-qPCR for viral transcripts (which would not be limited to available antibodies, and could be used in concert with multi-round infection, as in Figure 3B.)

The reviewers ask for a qRT-PCR analysis of viral gene expression in IRE1 knockout MEFs versus WT MEFs in a multistep infection experiment, similar to the growth curves in Figure 3B. Following this suggestion, we have determined the transcript levels of 4 viral genes over the course of 9 days. The results are shown in Figure 3—figure supplement 1. Consistent with the replication kinetics in Figure 3B, the transcript levels were massively reduced in IRE1 ko cells.

d) Some conclusions are not adequately supported by the data. For example, while the authors provide evidence for transient IRE1 activation during MCMV infection via an RT-qPCR assay that measures XBP1 mRNA splicing, they do not corroborate this data with more direct evidence of IRE1 activation via immunoblotting for IRE1 phosphorylation or XBP1s protein accumulation, which is standard practice in the UPR field. Moreover, they do not investigate whether MCMV infection (in either low MOI or high MOI infections) affects the levels or activation of IRE1, XBP1, or any other UPR component. In this regard, application of the RT-qPCR assay for XBP1 splicing to an MCMV infection time course would be quite informative.

In comments 1d and 1e (below), the reviewers ask that the qRT-PCR data in Figure 1 be accompanied by Western blot detection of IRE1 and XBP1s. We agree that this is a good way to support our claim that MCMV activates IRE1 and XBP1s at early times post infection. A Western blot time course showing total IRE1, phosphorylated IRE1, and XBP1s has been included in Figure 1 as a new panel C. The levels of pIRE1 and XBP1s are consistent with the spliced *Xbp1* transcript levels detected by qRT-PCR.

A qRT-PCR assay for XBP1 splicing over an extended MCMV infection time course has previously been published (see Stahl et al., 2013). The published time course had a focus on later time points and showed only a few early time points. Hence, the transient activation of Xbp1 splicing between 5 and 7 hpi, which we show here, was not detected in the previous paper.

e) It remains unclear whether MCMV infection causes alterations in the levels or activation of any of the cellular UPR machinery like IRE1, XBP1s or ATF6. At the least, this RT-qPCR data should be accompanied by western blots that show clear accumulation of XBP1s protein, so the authors can confidently conclude that IRE1 is indeed being activated and having the anticipated effect on XBP1s protein accumulation. The authors should directly demonstrate IRE1 activation in response to infection using conventional western blotting techniques, looking for IRE1 phosphorylation and impeded electrophoretic mobility, or accumulation of XBP1u or XBP1s proteins.

This comment is similar to comment 1d. Please see response to comment 1d above.

2) In Figure 2, it remains formally possible that higher doses of dox in Figure 2 negatively impact MCMV replication in an IRE1-independent manner. Importantly, this experiment lacks an empty vector control that would clarify this matter.

We agree that it remains formally possible that doxycycline and the reverse tetracycline transactivator (rtTA=TetON) negatively influence MCMV replication. To formally exclude this possibility, we used IRE1^-/-^ cells expressing TetON (the parental cells of the TetON IRE1-GFP cells in Figure 2) and WT MEFs transduced with a TetON-expressing lentivirus for multistep MCMV replication kinetics. As expected, doxycycline levels had no influence on MCMV replication (Figure 2—figure supplement 1).

3) The data are remarkably consistent between biological replicates. Are these true biological replicates performed at different times, or cells infected in parallel with the same virus stocks on the same day? Some clarity on this point, and a look at the primary data, would be illuminating.

Biological replicates were obtained by doing the entire experiment several times in parallel (example: growth curves). The entire experiment was done several times in parallel, not just the measurement (the latter would be a technical replicate). Biologically independent experiments were done separately on different days (e.g., Western blots). The source data files have been uploaded as supplementary data and are available to reviewers and readers.

4) This study would be significantly strengthened by identification of a viral factor responsible for IRE1 activation; this may be beyond the scope of the current study, but the reader would benefit from understanding approaches to identify such a factor among MCMV early gene products.

We fully agree that it would be interesting to know which viral factor(s) is/are responsible for IRE1 activation, and we also agree that this is beyond the scope of the current study. We do not know yet which viral proteins are responsible, but there are a few obvious candidates. Calcium release from the ER can cause ER stress and activate IRE1, and therefore viral protein inducing ER calcium release would be good candidates. In HCMV, the UL37x1 protein causes ER calcium release (Sharon-Friling et al., 2006). Its homolog in MCMV, the m38.5 protein, might have the same function. Moreover, MCMV encodes several G protein-coupled receptors (GPCRs). GPCR-dependent activation of phospholipase C and production of IP3 can also induce ER calcium release. Indeed, preliminary experiments in our lab have shown that inhibitors of PLC or the IP3 receptor inhibit the MCMV-induced Xbp1 mRNA splicing.

The transient induction of Xbp1 splicing occurs at early times (peak at 6 hpi), whereas massive accumulation of MCMV glycoproteins occurs much later. Most of the viral envelope glycoproteins are expressed at late times post infection (i.e., starting 12 hpi). However, it is possible that certain early glycoproteins are particularly potent ER stress inducers, either by causing ER calcium release (as suggested above) or by binding and sequestering BiP. In the revised manuscript, we discuss possible mechanisms (Discussion paragraph one).